# Validity of the Computerized Battery for Neuropsychological Evaluation of Children (BENCI) in Spanish Children: Preliminary Results

**Manuel Fernández-Alcántara** [1], **Natalia Albaladejo-Blázquez** [1], **María Inmaculada Fernández-Ávalos** [1], **Miriam Sánchez-SanSegundo** [1], **Francisco Cruz-Quintana** [2], **Vanesa Pérez-Martínez** [3], **Claudia Carrasco-Sánchez** [2] and **María Nieves Pérez-Marfil** [2,*]

[1] Department of Health Psychology, University of Alicante, 03690 Alicante, Spain; mfernandeza@ua.es (M.F.-A.); natalia.albaladejo@ua.es (N.A.-B.); inmaculada.fernandez@ua.es (M.I.F.-Á.); miriam.sanchez@ua.es (M.S.-S.)
[2] Mind, Brain and Behaviour Research Centre (CIMCYC), University of Granada, 18071 Granada, Spain; fcruz@ugr.es (F.C.-Q.); claudiacs@correo.ugr.es (C.C.-S.)
[3] Department of Community Nursing, Preventive Medicine and Public Health and History of Science, University of Alicante, 03690 Alicante, Spain; vanesa.perez@ua.es
* Correspondence: nperez@ugr.es

**Abstract:** Study of the neurodevelopment of children is vital to promote good quality of life during childhood. Few batteries showing adequate reliability and validity indices are available to evaluate the different neuropsychological domains. The objective of this study was to obtain initial evidence on the validity of the Computerized Battery for Neuropsychological Evaluation of Children (BENCI) in a Spanish population. To assess the validity of the BENCI battery and other measures of task switching, abstract reasoning, linguistic abilities, processing speed, and attention were used. The sample was composed of a total of 73 children aged 9, 10, and 11 years. Significant differences among age groups were observed in the domains of sustained attention, memory, and executive function. In addition, the BENCI subtests showed statistically significant correlations with the other neuropsychological tools. Further research is warranted on the relationship of the BENCI with other tests in wider age groups and to assess the factorial structure of the scale and the reliability values of the subtests. In conclusion, this study seems to indicate that the Spanish version of the BENCI has promising validity to be used for evaluating the main neuropsychological domains in children.

**Keywords:** neuropsychology; executive function; attention; memory; assessment; children

## 1. Introduction

The main objective of child psychology is to evaluate and study the cognitive, emotional, and behavioral development of children's brains. One of the main limitations in this field is the scarcity of instruments for evaluating different neuropsychological domains to detect potential problems and to determine the effectiveness of interventions [1].

Over the past few years, neuropsychological evaluation batteries have been developed to obtain profiles of most neuropsychological domains, including a computerized battery for children (*Batería Computerizada de Evaluación Neuropsicológica Infantil*, BENCI) developed through an international development project involving Spain, Morocco, Argentina, and Ecuador [2]. The BENCI is based on the most widely accepted models of childhood neuropsychological development [3] and comprises numerous neuropsychological tests. It evaluates the neuropsychological domains of processing rate, visual–motor coordination, attention, memory, language, and executive function. The tests can be completed quickly and are especially designed to be attractive to children. The BENCI is freely available in tablet format, and both the test application and the data gathering are performed with this device, maximizing the reliability of results.

The BENCI has been adapted and validated for Arab [4] and Ecuadorian [2] populations. For the former, intraclass correlation coefficients ranged between 0.51 and 0.81, indicating adequate reliability, and the internal structure of the battery also had acceptable fitting indexes for a five-factor model: inhibition, flexibility, fluidity, reasoning, and verbal memory (CFI = 0.939, TLI = 0.942, RMSEA = 0.042). In the Ecuadorian version, test–retest correlations were determined as a measure of reliability, obtaining statistically significant values between 0.97 and 0.35. Additionally, correlations with other neuropsychological tests were examined to assess the convergent validity and yielded values between 0.33 and 0.69, indicating a close correlation with tests evaluating the same domains [2]. The psychometric values obtained for the BENCI are in line with those reported for other neuropsychological batteries used in adults [5,6].

With respect to discriminant validity, various studies have shown that the BENCI performance varies in terms of the age and population in which it is applied. In a study of 274 Ecuadorian children, the BENCI showed differences in neuropsychological performance as a function of the socioeconomic status (SES) of the family. Children with low SES showed worse performance than children with medium SES in most neuropsychological domains studied. These results follow the same pattern as the evidence gathered in studies that took place in Madagascar [7], Argentina [8], Brazil [9], Canada [10], and the United States [11]. Although there is compelling evidence that suggests an impact of a low SES on the child's emotional, social, and cognitive development, studies on the impact on the neuropsychological development remain scarce and not very culturally diverse. As previously mentioned, a standardized battery like the BENCI could help to improve research on the topics. In addition, interactions were found between SES and age in the following domains: executive function, verbal memory, and language [12]. In clinical populations, the BENCI has demonstrated adequate discriminant validity for neuropsychological performance between premature and full-term neonates at the age of 7 years [13]. Particular BENCI tests (e.g., visual memory) also proved to discriminate performance between children with and without learning difficulties at different ages [14].

BENCI scores appear to be sensitive to specific neuropsychological interventions. Children aged 4 years who received an executive function stimulation program showed improvements in attention, visual perception, and executive function, as evaluated with BENCI and the CUMANIN (Childhood Neuropsychological Maturity Questionnaire) and BRIEF-P (Behavioral Evaluation of Executive Functioning–Children's Version) instruments [15].

However, the validity of the BENCI battery has not yet been studied in a Spanish population. In the clinical setting, the availability of validated child neuropsychological evaluation instruments in the mother tongue allows potential child neurodevelopment problems to be detected in a rapid manner, reducing their impact on the health-related quality of life of the children and their families.

With this background, the objective of the present study was to obtain preliminary evidence on the validity of the BENCI in a Spanish population, exploring its discriminant validity by comparing scores among different age groups and examining its convergent validity by comparing scores with the results of neuropsychological tests habitually employed to evaluate children. The hypothesis were: (1) BENCI tasks will discriminate between age groups, with older children having better scores than younger children, and (2) positive correlations will be found between BENCI tasks and other neuropsychological test that assess the same domain (task switching, abstract reasoning, linguistic abilities, processing speed, and attention).

## 2. Materials and Methods

### 2.1. Participants and Procedure

We selected 73 children from three public schools in the province of Alicante (Spain). The final sample included 16 boys and 12 girls in the fourth grade of primary school, 11 boys and 14 girls in the fifth grade, and 8 boys and 12 girls in the sixth grade. The mean age was 9.89 years (SD = 0.81). The mean weight of the children was 42.61 kg (SD = 10.67),

with a mean height of 149.71 cm (SD = 6.88) and mean head circumference of 58.12 cm (SD = 2.13). Most children (57.5%) had one sibling, 23.3% had none, 12.3% had two siblings, and 6.5% three or more siblings; 95.9% of children were right-handed. Inclusion criteria were school children aged between 9 and 11 years and able to read and write. Exclusion criteria were mother tongue other than Spanish, sensory or cognitive deficits, previous diagnosis of learning difficulties, and high levels of depression and/or anxiety.

Tests were conducted by two examiners (M.I.F.-A. and V.P.-M.) with wide experience in child neuropsychological evaluation and 12 months of training in BENCI administration. Tests took place in the school, using a room with adequate physical conditions. The evaluation took around 90 min, including the BENCI and the other neuropsychological tests. Before the evaluations, parents/guardians of all participating children received information on the study and signed their informed consent. The study was approved by the Ethics Committee of the University of Alicante (Ref. UA-2019-03-15) and the Education, Research, Culture, and Sports Ministry of the Community of Valencia.

### 2.2. Measures

*Sociodemographic variables*: Data were gathered on the age, sex, number of siblings, weight, height, head circumference, and laterality of the children.

*Computerized Battery for Neuropsychological Evaluation of Children—BENCI* [2].

The BENCI is a computerized battery developed through the International Cooperation for Development Programs in Ecuador, Morocco, and Argentina (References: A3/042954/11, PE18X and P1181X). The original version of the BENCI battery [2] was developed in Spanish among Ecuadorian children between 6 and 11 years old. The Arabic version of the BENCI battery is also available [4].

The BENCI included the basic neuropsychological domains required to conduct a complete neuropsychological assessment [16]: attention, memory, speed processing, visuomotor coordination, language, and executive functions (see Table 1).

**Table 1.** Main domains, areas and tasks that compose the BENCI.

| Domain | Area | BENCI Task |
|---|---|---|
| Processing Speed | | Simple reaction time test |
| Visuomotor Coordination | | Visuomotor |
| Attention | Sustained<br>Selective | Continuous performance task (CPT)<br>Spatial Stroop |
| Memory | Verbal<br>Visual | Verbal memory test<br>Visual memory test |
| Language | Comprehension<br>Production | Verbal comprehension (images and figures)<br>Semantic and phonetic fluency |
| Executive Function | Updating<br><br><br>Inhibition<br>Flexibility<br><br>Planning | Working memory<br>Abstract reasoning<br>Semantic fluency<br>Go/no-go<br>Alternate visouomotor<br>Spatial Stroop<br>Amusement park |

A set of neuropsychological tests were developed to assess these domains. The different tests were developed using valid neuropsychological procedures based on previous neuropsychological assessment literature [16]. It offers comprehensive neuropsychological evaluation of the following neuropsychological domains [4,12]: processing speed, visual–motor and alternate visual–motor coordination, attention, language (phonetic fluency and comprehension), memory (verbal and visual memory), and executive functions (inhibition, abstract reasoning, planning, semantic fluency, and working memory). The battery is in tablet format (iPad, 9.7″ screen), allowing standardized administration and reliable record-

ing of correct answers (CAs), errors, and/or reaction time (RT) for every test. The BENCI is easy to use and designed to be attractive to children. The battery requires 60–70 min for completion, and there is a 10 min rest period halfway through the session. The initial study in Morocco identified a factorial structure for the components of executive function of five factors: inhibition, flexibility, verbal fluency, reasoning, and memory [4]. The recommendations of Lezak et al. [16] were followed, and the order of test administration was the same for all participants. The domains and tests are described below.

*Processing Speed*.

Test 1. Simple reaction time test. Consists of pressing any key, as fast as possible, each time a cross is displayed on the screen (+). Parameters recorded: reaction time (RT) (ms).

*Visuomotor Coordination*.

Test 2. Visuomotor. It is composed of two tasks: simple and alternate. The simple task consists of touching, in an increasing order or according to a given sequence, the numbers or elements that appear randomly on the screen. The alternate visuomotor task consists of pressing, in alternating and ascending order, the numbers of two different series that are displayed randomly on the screen. Parameters recorded: RT (ms).

*Sustained Attention*.

Test 3. Continuous performance test (CPT). Blocks of letters (trails) are displayed on the screen, one after the other. Participants must press a key every time the correct stimulus appears (e.g., an A after an X). All other letters are distractors. Parameters recorded: RT (ms) and number of correct responses.

*Episodic Memory*.

Test 4. Verbal memory. The participant listens to the same series of words three times and, at the end of each sequence, must repeat aloud all the words he/she remembers. Parameters recorded: correct responses.

Verbal memory (delayed and recognition trails). In the delayed test, 20 min after the end of the verbal memory test, the participant must repeat aloud all the words he/she remembers from the list that was presented in that test. In the recognition test, immediately after the previous test, the participant listens to a group of words, half of which belong to the list presented in the verbal memory test, and must state whether each of them was on that list. Parameters recorded: correct responses.

Test 5. Visual memory. Drawings of common objects are presented and the participant must then repeat aloud all the drawings he/she remembers. Parameters recorded: correct responses.

Visual memory (delayed and recognition trails). In the delayed test, 20 min after the end of the visual memory test, the child has to repeat aloud all the pictures he/she remembers from the list that was presented in the test. In the recognition test, immediately after the previous test, images are presented, many of which appeared in the visual memory test, and for each of them the person must indicate whether it was in the sequence of the mentioned task. Parameters recorded: correct answers.

*Language*.

Test 6. Verbal comprehension (images). A set of pictures from a given category (e.g., animals) is shown and the participant is instructed (orally) to select a picture that fulfils the indicated conditions (type of animal, position, activity it can perform, and/or color: e.g., "Touch the frog next to the dog"). Parameters recorded: correct answers.

Test 7. Verbal comprehension (figures). Similar to the previous test, but with pictures of geometric figures (circles, triangles, and squares: small, medium, and large) of different colors, pointing out those that meet the indicated conditions (figure, size, position, and/or color: e.g., "Touch a small blue circle"). Parameters recorded: correct answers.

Test 8. Phonetic fluency. A letter is indicated and the participant has to mention all the words he/she knows that begin with that letter. Parameters recorded: correct answers. Time: 60 s.

*Executive Function*.

Test 9. Working memory. After hearing sequences of mixed numbers and colors, the participant has to repeat the numbers and colors stated; first the numbers, in ascending order, and then the colors, or vice versa. Parameters recorded: correct answers.

Test 10. Abstract reasoning. A set of logical series is presented on screen. The participant must select the element that completes the series currently displayed. Parameters recorded: RT (ms) and correct answers.

Test 11. Semantic fluency. The participant listens to a semantic category (colors or animals) and must say all the items he/she knows from that category. Parameters recorded: correct answers. Time: 60 s.

Test 12. Inhibition: Go–No go. Two items are alternately displayed on the screen. In the first phase, the participant must press a key when one of them appears, and, after hearing a sound, which represents the change to the second phase or half of the test, press a key when the other item appears. Parameters recorded: RT (ms) and correct responses.

Test 13. Flexibility and inhibitory control: Spatial Stroop. A sequence of arrows is presented on the screen that can point to the left or to the right. Each time an arrow appears, the participant must press the LEFT ARROW key if the arrow is pointing to the left and the RIGHT ARROW key if the arrow is pointing to the right. On some occasions, the stimuli are presented on the side of the screen consistent with the direction of the arrow, and on other occasions, the opposite. Parameters recorded: RT (ms), correct answers, and errors by omission and commission.

Test 14. Planning. An amusement park is presented and the role of the participant is to go on the maximum number of rides with the money provided. He/she will have to bear in mind that there is a time limit for being at the funfair and each ride takes up a part of that time. The objective of the task is that the participant manages to go on the highest number of different rides with the resources available. Parameters recorded: number of rides and number of different rides used.

*Trail-Making Test (TMT)* [4,17].

This test measures visual attention and task switching. The paper-and-pencil test lasts a maximum of five minutes and has two parts [17]. In the first part, numbers from 1 to 25 are displayed on the sheet in random order. The child must connect the numbers in consecutive order, without lifting the pencil from the paper. In part B, numbers from 1 to 13 are presented with the first 12 letters of the alphabet, having to alternately match letters and numbers. No scale is available for the child population in Spain; therefore, the time taken to carry out parts A and B was recorded in seconds. There is no need to continue the test after five minutes have passed [4].

*Toni 2* [18,19].

This non-verbal intelligence test was developed by Brown et al. and lasts 15–20 min. It evaluates intelligence without using verbal material, based on the ability to solve abstract tests. Neither the evaluator nor the child reads, writes, listens, or speaks during the test. It includes two equivalent forms (A and B). The forms are presented in increasing order of difficulty (55 variations). Each time, a drawing with one incomplete part is presented along with a series of alternative parts, and the participant must choose the element that completes the drawing.

*Token Test* [20].

The Token test evaluates linguistic abilities and contains 62 instructions of gradually increasing difficulty. Twenty tokens are used, being square or circle in shape, large or small, and in five different colors. The 62 instructions are divided into five parts. Scores are added together to obtain the test result.

*Stroop Task* [21,22].

The Stroop paradigm evaluates attention and its relationship with automatisms and also measures inhibition. We applied the Spanish adaptation [22], which uses three sheets containing 100 items each. The first sheet has the words "red", "green", and "blue" written in black ink. The child must read aloud as many of the words as possible. The second sheet displays groups of four Xs ("XXXX"), with each group being colored red, green, or blue,

and the child must name the ink color of as many groups as possible. Finally, the third sheet bears the names of colors written in a color that does not match the meaning (e.g., "blue" written in red ink); the child must name the color of the ink, ignoring the meaning of the word. The complete test lasts 5 min. The RT was recorded for each task.

*WISC IV Animals Sub-Test* [23].

This test evaluates processing speed, selective attention, and planning. In part A, animals appear in a disorganized manner among other elements, and the child has 45 s to cross out all of the animals they detect. In the second part, animals are ordered in rows and the child has another 45 s to select them. The score is obtained as the difference between correct and incorrect answers.

*Revised Child Anxiety and Depression Scale (RCADS)—Anxiety* [24].

This instrument is used to evaluate the symptomatology of the main anxiety disorders. It presents six statements related to DSM-IV criteria with 4-point Likert-type responses (0–3). In the present study a T score $\geq$ 65 was considered to indicate elevated anxiety.

*Child Depression Inventory (CDI)—Depression* [25].

CDI is one of the most widely accepted instruments for evaluating childhood depression. It includes 27 items, which are each presented with three sentences on their intensity or frequency (i.e., a total of 81 elements). They are related to most criteria for a child depression diagnosis. It has two scales: dysphoria (depressive mood, sadness, concern, etc.) and negative self-esteem (judgments of inefficiency, ugliness, evilness, etc.). A total depression score is obtained by adding the values for each item together, with a higher score indicating higher intensity or frequency of depressive symptoms. The questionnaire takes 10–15 min to apply. The cutoff score for exclusion from the present study was 19 points.

### 2.3. Data Analysis

Mean scores were calculated for all tests (BENCI and other neuropsychological tests). In the validity analysis, performance in the tests was compared as a function of the age of the children, conducting ANOVAs with age as a three-level independent variable and the BENCI score as the dependent variable. *Post-hoc* effects were studied using the Bonferroni correction. Finally, non-parametric correlations (Spearman's *rho*) were tested between specific BENCI tests and tests evaluating similar domains. $p < 0.05$ was considered significant.

## 3. Results

### 3.1. BENCI Scores and Discriminant Validity

Table 2 lists the mean scores for each neuropsychological test used in the present research. Tables 3 and 4 exhibit the mean score for each BENCI test by age.

**Table 2.** Descriptive statistics of the neuropsychological tests.

| Test | Mean (SD) |
|---|---|
| TMT–A (RT) | 48.51 (17.41) |
| TMT–B (RT) | 110 (42.10) |
| Toni 2 (DS) | 24.78 (7.91) |
| Token (CAs) | 33.19 (2.44) |
| Stroop Condition 1 (CAs) | 82.33 (10.76) |
| Stroop Condition 2 (CAs) | 54.66 (11.20) |
| Stroop Condition 3 (CAs) | 32.71 (6.94) |
| Animals Form A (CAs) | 34.52 (9.71) |
| Animals Form B (CAs) | 41.15 (9.62) |
| RCADS (DS) | 7.37 (3.08) |
| CDI (DS) | 6.78 (4.59) |

Note. TMT = Trail Making, Test, RCADS = Revised Child Anxiety and Depression Scale, CDI = Child Depression Inventory, RT = reaction time, Cas = correct answers, DS = direct score, SD = standard deviation.

**Table 3.** Effects of age on processing speed, visual–motor coordination, sustained attention, and memory tests (*Batería de Evaluación Neuropsicológica Computerizada Infantil* (BENCI)).

| Test | 9 Years n = 28 M (SD) | 10 Years n = 25 M (SD) | 11 Years n = 20 M (SD) | F | p | Post-Hoc |
|---|---|---|---|---|---|---|
| **Simple reaction test** | | | | | | |
| Median (RT) | 484.11 (57.73) | 458.68 (131.46) | 451.90 (60.22) | 0.883 | 0.418 | |
| **Visual–motor coordination** | | | | | | |
| Visual–motor (RT) | 26.69 (7.57) | 27.64 (8.88) | 26.42 (9.72) | 0.131 | 0.878 | |
| Alternate visual–motor (RT) | 42.09 (14.63) | 45.58 (15.47) | 40.69 (13.07) | 0.698 | 0.501 | |
| **Sustained attention** | | | | | | |
| CPT (CAs) | 56.82 (2.50) | 58.12 (1.83) | 58.50 (1.96) | 4.218 | 0.019 | 9 = 10 10 = 11 9 < 11 |
| **Memory** | | | | | | |
| Verbal memory. E1 (CAs) | 4.54 (0.88) | 4.80 (0.82) | 4.85 (0.93) | 0.945 | 0.394 | |
| Verbal memory. E2 (CAs) | 5.32 (0.86) | 5.64 (0.49) | 5.50 (0.51) | 1.533 | 0.223 | |
| Verbal memory. E3 (CAs) | 5.68 (0.55) | 5.80 (0.41) | 5.75 (0.44) | 0.436 | 0.648 | |
| Delayed verbal memory (CAs) | 4.86 (1.33) | 5.36 (0.76) | 5.20 (0.89) | 1.604 | 0.208 | |
| Recognition verbal memory (CAs) | 11.25 (0.80) | 11.80 (0.41) | 11.10 (0.79) | 6.809 | 0.002 | 9 < 10 9 = 11 10 > 11 |
| Visual memory (CAs) | 7.82 (2.07) | 8.12 (1.74) | 9.15 (2.23) | 2.676 | 0.076 | |
| Delayed visual memory (CAs) | 6.36 (2.02) | 6.36 (1.58) | 7.10 (2.36) | 1.013 | 0.369 | |
| Recognition visual memory (CAs) | 45.96 (2.40) | 47.16 (1.99) | 45.05 (3.50) | 3.666 | 0.031 | 9 = 10 9 = 11 10 > 11 |

Note: CPT = continuous performance task, RT = reaction time, Cas = correct answers.

**Table 4.** Effects of age on language and executive function tests (*Batería de Evaluación Neuropsicológica Computerizada Infantil* (BENCI)).

| Test | 9 Years n = 28 M (SD) | 10 Years n = 25 M (SD) | 11 Years n = 20 M (SD) | F | p | Post-Hoc |
|---|---|---|---|---|---|---|
| **Language** | | | | | | |
| Verbal comprehension (images) (CAs) | 9.54 (0.69) | 9.80 (0.41) | 9.60 (0.60) | 1.437 | 0.245 | |
| Verbal comprehension (shapes) (CAs) | 9.32 (0.67) | 9.56 (0.71) | 9.50 (0.69) | 0.858 | 0.429 | |
| Phonetic fluency (CAs) | 6.07 (2.42) | 6.80 (3.40) | 7.50 (3.25) | 0.514 | 0.600 | |
| **Executive function** | | | | | | |
| Working memory (CAs) | 7.32 (1.91) | 7.56 (2.14) | 8.40 (2.23) | 1.652 | 0.199 | |
| Abstract reasoning (CAs) | 16.61 (5.13) | 18.12 (3.32) | 20.30 (3.13) | 4.795 | 0.011 | 9 = 10 10 = 11 9 < 11 |
| Semantic fluency (CA) | 15.18 (4.72) | 17.56 (3.49) | 17.50 (4.81) | 2.513 | 0.088 | |
| Inhibition 1 (go–no-go) (CAs) | 82.57 (8.75) | 84.60 (9.60) | 91.13 (8.34) | 5.596 | 0.006 | 9 = 10 10 = 11 9 < 11 |
| Inhibition 2 (go–no-go) (CAs) | 82.54 (13.61) | 90.15 (11.58) | 92.27 (8.79) | 4.742 | 0.012 | 9 = 10 10 = 11 9 < 11 |
| Inhibition 3 (go–no-go) (CAs) | 75.55 (11.47) | 82.95 (9.40) | 85.53 (9.40) | 6.084 | 0.004 | 9 < (10 = 11) |
| Inhibition 4 (go–no-go) (CAs) | 81.91 (13.14) | 91.19 (8.68) | 90.20 (10.39) | 5.583 | 0.006 | 9 < (10 = 11) |
| Planning (rides visited) | 10.86 (1.92) | 11.20 (1.63) | 11.10 (1.02) | 0.315 | 0.731 | |
| Planning (different rides) | 8.32 (0.82) | 8.40 (0.87) | 8.75 (0.55) | 1.929 | 0.153 | |

Note: RT = reaction time, CAs = correct answers.

Tables 3 and 4 also display ANOVA results for discriminant validity as a function of age (9, 10, and 11 years). As shown in Table 3, significant differences in sustained attention were observed between the 7- and 9-year-olds (*p* < 0.019), with better performance in the

older children, whereas significant differences in verbal and visual recognition memory tests were found between the group of 10-year-lds and the 9- and 11-year-olds, with the 10-year-olds recalling a larger number of words in comparison to the other two age groups (see Table 3). Finally, with regard to executive function (Table 4), statistically significant differences were observed between the 9- and 11-year-olds in abstract reasoning and in the number of correct answers in the inhibition test, with higher scores observed for the older children (see Table 4). Improved performance was observed in many other tests, but statistical significance was not reached (see Tables 3 and 4).

### 3.2. Convergent Validity of the BENCI

Spearman correlations showed associations between the neuropsychological tests used and the BENCI (see Table 5). RTs for TMT A (*rho* = 0.330) and B (*rho* = 0.292) were positively associated with visual–motor test results. The sustained attention test was positively correlated with condition 3 of Stroop (*rho* = 0.274) and the total Token test score (*rho* = 0.317) was negatively correlated with the RT of TMT B (*rho* = −0.243). CAs in the visual memory task were positively associated with the total Token test score (*rho* = 0.314) and with animal test A (*rho* = 0.336) and B (*rho* = 0.250) scores. Positive associations were found between the verbal comprehension test and Token test scores (*rho* = 0.250), and between the semantic fluency and Token test scores (*rho* = 0.424). Finally, a significant correlation was observed between the abstract reasoning test and Toni 2 scores (*rho* = 0.569).

**Table 5.** Correlations between BENCI tests and other neuropsychological tests.

| BENCI Test | Neuropsychological Test | *Rho* |
|---|---|---|
| Visual–motor (RT) | TMT A | 0.330 ** |
| | TMT B | 0.292 * |
| CPT (CAs) | Stroop (condition 3) | 0.274 * |
| | TMT B | −0.243 * |
| | Token | 0.317 ** |
| Visual memory (CAs) | Token | 0.314 ** |
| | Animals A | 0.336 ** |
| | Animals B | 0.250 * |
| Verbal comprehension (images) (CAs) | Token | 0.238 * |
| Semantic fluency (CAs) | Token | 0.424 ** |
| Abstract reasoning (CAs) | Toni 2 | 0.569 ** |

Note: TMT = Trail Making Test, CPT = continuous performance task, RT = reaction time, CAs = correct answers ** $p < 0.01$, * $p < 0.05$.

### 4. Discussion

The objective of this study was to obtain preliminary evidence on the discriminant and convergent validity of the BENCI in a Spanish population. To our knowledge, this is the first study to provide normative values for the BENCI in a Spanish population and the first to analyze its relationship with other neuropsychological evaluation instruments. The BENCI is designed to identify potential and actual neurodevelopment deficits and can be used to assess the effects of interventions by detecting changes in the corresponding functions. It allows for evaluation of the main neuropsychological domains in a structured manner and is useful in both clinical and research settings. The results suggest that the BENCI can discriminate among children of different ages and that their scores correlate with those obtained using conventional neuropsychological tests for similar domains.

Childhood is the most important stage of human development, as it is in the first years of life that the maturational and neurological foundations of development are laid. What, how, and how much children learn in school depends largely on the social, emotional, and cognitive competencies they have developed in their early years. Comprehensive instruments to assess children's neuropsychological performance are needed to identify children with possible deficits early, as well as to evaluate the effect of early intervention and prevention programs. The identification or diagnosis of potentially affected neurodevelopmental processes is one

of the basic prevention activities. Neuropsychological assessment in children has specific characteristics, since functional systems of the brain are assessed when development is not yet complete. Neuropsychological assessment of children requires specific tests to identify the child's performance in different neuropsychological domains. Some tests exist but, for the most part, they do not allow an initial screening and do not identify specific areas of deficit, but offer general scores and are expensive, which makes their use difficult, even more so in certain contexts. Taking these difficulties into account, it is necessary to have assessment instruments that are flexible, that can be modified according to age, that are attractive and quick to apply, and that have been developed specifically for children.

BENCI scores were previously reported to vary among age groups, but the differences in age were more extreme [4,12,14]. In the present study, there was a tendency for younger children to score lower in most tests in the battery. Various longitudinal studies have shown the age between 7 and 11 years to be a critical period for the development of executive function and other neuropsychological domains [3,26,27].

The convergent validity findings for the BENCI are in line with the first results published for this battery. A pilot study using the initial version, which evaluated 43 children aged between 6 and 11 years, found associations between the Children's Color Trails Test and visual–motor/alternate visual–motor tests, between selective attention (CPT) and Stroop, and between the reasoning test and Raven's progressive matrices [2]. The present study replicated these findings for similar tests and obtained additional evidence, using neuropsychological tasks such as the Token test, Animals test, and TMT. In addition, anxiety and depression findings were used as exclusion criteria.

TMT has conventionally been used to evaluate attention, processing speed, and visual–motor coordination, and it has been associated with the results of visual–motor and sustained attention tests [28,29]. Although the Token test was initially applied to detect language problems, its scores have been associated with other neuropsychological functions, including memory and attention processes [30], which may explain the association found between the results for this test and those for sustained attention and visual memory. The association between the Token test findings and those for the verbal comprehension of images reflects similarities between them in the stimuli used and domains investigated (comprehension of complex orders). Finally, Token test results are related to those for semantic fluency test because linguistic ability is important in both tests [1].

This study has various limitations, including the narrow age range, which may explain the absence of statistically significant age differences in some domains. In addition, the IQ of the children was not controlled for in the analyses. However, strict selection criteria were imposed to ensure that the children had no neuropsychological or affective problems, ensuring adequate data for the validity study. Research in larger samples and wider age ranges are required to develop score curves, and longitudinal studies are warranted to explore test–retest correlations and further verify the reliability of this battery.

## 5. Conclusions

In conclusion, this study demonstrates that the Spanish version of the BENCI has adequate validity to be used for evaluating the main neuropsychological domains in children. Adequate discriminant validity, showing that older children tend to have higher scores in some of the BENCI tests, was shown. In addition, positive and significant correlations were found between neuropsychological tests and the BENCI tests that targeted the same neuropsychological domain. These results should be taken with caution, and further research is needed with larger sample sizes that allows for testing the factorial structure of the BENCI in a Spanish population of children and adolescents.

**Author Contributions:** Conceptualization, F.C.-Q. and M.N.P.-M.; data curation, M.F.-A., M.I.F.-Á. and V.P.-M.; formal analysis, M.I.F.-Á., F.C.-Q. and M.N.P.-M.; funding acquisition, M.F.-A. and M.N.P.-M.; methodology, N.A.-B. and M.S.-S.; project administration, M.F.-A. and M.N.P.-M.; resources, N.A.-B.; software, M.F.-A. and V.P.-M.; supervision, M.S.-S. and F.C.-Q.; validation, M.I.F.-Á.; writing—original draft, M.F.-A. and M.N.P.-M.; writing—review and editing, M.F.-A., N.A.-B., M.I.F.-Á., M.S.-S.,

F.C.-Q., V.P.-M., C.C.-S. and M.N.P.-M. All authors have read and agreed to the published version of the manuscript.

**Funding:** This study was funded by the *Conselleria d'Educació, Investigació, Cultura i Esport* de la Generalitat Valenciana (Proyectos I+D+i desarrollados por grupos de investigación emergentes) (GV/2017/166) (PI: Manuel Fernández-Alcántara) and by the *Agencia Andaluza de Cooperación Internacional al Desarrollo de la Junta de Andalucía* (Proyectos de Investigación para la Cooperación Internacional al Desarrollo AACID-Universidades) (Ref. 2020U1006) (PI: María Nieves Pérez Marfil).

**Institutional Review Board Statement:** The study was conducted in accordance with the Declaration of Helsinki, and approved by Ethics Committees of the University of Alicante (Ref. UA-2019-03-15), University of Granada (Ref. 945/CEIH/2019), and the Department of Education of Education of the communities of Valencia and Andalusia in Spain.

**Informed Consent Statement:** Informed consent was obtained from all subjects involved in the study.

**Data Availability Statement:** The data that support the findings of this study are available from the corresponding author, upon reasonable request.

**Acknowledgments:** The authors thank all students who devoted their time to this study, and to the schools that collaborated in the research.

**Conflicts of Interest:** The authors declare no conflict of interest.

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
