# Peer review of "Validity of the Computerized Battery for Neuropsychological Evaluation of Children (BENCI) in Spanish Children: Preliminary Results"

_ejihpe, doi:10.3390/ejihpe12080065_

Round 1

Reviewer 1 Report

Review of Validity of the Computerized Battery for Neuropsychological 2

Evaluation of Children (BENCI) in Spanish children: Preliminary Results

Overall evaluation: I found the paper an interesting read but am unfortunately not in the position to evaluate it properly as only the (pretty standard) tasks used to assess external validity are described and none of the BENCI task which I consider essential to evaluate the paper properly. If those tasks are structurally similar to the EF tasks used, then the high correlations are no surprise but this not judgeable from the present manuscript. Thus, please include a description of the BENCI task.

Finally, I wondered who did translate the tasks, please clarify. Why were the tasks chosen as they were, please motivate (maybe some former study did use the exact same tasks in a different sample, maybe those tasks measure abilities most critical for successful neurotypical development or pathologies).

Next, given the long introduction on the impact of SES on EF performance and neurotypical development I wondered why the scores were analysed only including age and not SES, maybe the sample was to homogenous SES-wise. In addition, the five-factor structure of the BENCI is also mentioned quite often without reference or fit indices which is deficient. So please provide at least the fit indices. However, this brings me to my next point, how was sample size determined and which age groups would be more informative (as suggested in the GD). Seventy-three seems a bit too small sample for a CFA. Please convince me otherwise.

Thus, overall, I agree that test batteries are important, but I think they should be properly documented when validation studies are published. Thus, in a revision much more information needs to be provided to motivate readers to use the BENCI.

Author Response

Reviewer 1

Review of Validity of the Computerized Battery for Neuropsychological Evaluation of Children (BENCI) in Spanish children: Preliminary Results

  1. Overall evaluation: I found the paper an interesting read but am unfortunately not in the position to evaluate it properly as only the (pretty standard) tasks used to assess external validity are described and none of the BENCI task which I consider essential to evaluate the paper properly. If those tasks are structurally similar to the EF tasks used, then the high correlations are no surprise but this not judgeable from the present manuscript. Thus, please include a description of the BENCI task.

We thank the reviewer for their remark. We have added descriptive information of the tasks that compose the BENCI Battery, indicating the main differences and similarities with other neuropsychological tests.

Computerized Battery for Neuropsychological Evaluation of Children-BENCI [2].

BENCI is a computerized battery developed through the International Cooperation for Development Programs in Ecuador, Morocco, and Argentina (References: A3/042954/11, PE18X and P1181X). The original version of the BENCI battery [2] was developed in Spanish among Ecuadorian children between 6 and 11 years old. The Arabic version of the BENCI battery is also available [4].

BENCI included the basic neuropsychological domains required to conduct a com-plete neuropsychological assessment [16]: attention, memory, speed processing, visuo-motor coordination, language, and executive functions (see Table 1).

Table 1. Main domains, areas and tasks that compose the BENCI

Domain

Area

BENCI Task

Processing Speed

Simple reaction time test

Visuomotor Coordination

Visuomotor

Attention

Sustained

Selective

Continuous Performance Task (CPT)

Spatial Stroop

Memory

Verbal

Visual

Verbal Memory Test

Visual Memory Test

Language

Comprehension

Production

Verbal Comprehension (Images and Figures)

Semantic and Phonetic Fluency

Executive Function

Updating

Inhibition

Flexibility

Planning

Working Memory

Abstract Reasoning

Semantic Fluency

Go/No-Go

Alternate visouomotor

Spatial Stroop

Amusement Park

A set of neuropsychological tests were developed to assess these domains. The dif-ferent tests were developed using valid neuropsychological procedures based on previous neuropsychological assessment literature [16].  It offers comprehensive neuropsy-chological evaluation of the following neuropsychological domains [4,12]: processing speed, visual-motor and alternate visual-motor coordination, attention, language (pho-netic fluency and comprehension), memory (verbal and visual memory), and executive functions (inhibition, abstract reasoning, planning, semantic fluency and working memory). The battery is in tablet format (iPad, 9.7” screen), allowing standardized ad-ministration and reliable recording of correct answers (CAs), errors, and/or reaction time (RT) for every test. BENCI is easy to use and designed to be attractive to children. The battery requires 60-70 min for completion, and there is a 10-min rest period halfway through the session. The initial study in Morocco identified a factorial structure for the components of executive function of five factors: inhibition, flexibility, verbal fluency, reasoning, and memory [4]. Recommendations of Lezak et al. [16] are followed, and the order of test administration is the same for all participants. The domains and tests are described below.

Processing speed

Test 1. Simple reaction time test. Consists of pressing any key, as fast as possible, each time a cross is displayed on the screen (+). Parameters recorded: Reaction Time (TR) (ms).

Visuomotor coordination

Test 2. Visuomotor. It is composed by two tasks: Simple and Alternate. The simple consists of touching, in an increasing order or according to a given sequence, the numbers or elements that appear randomly on the screen. The alternate visuomotor task consists of pressing, in alternating and ascending order, the numbers of two different series that are displayed randomly on the screen. Parameters recorded: T.R. (ms).

Sustained attention

Test 3 .Continuous performance test (CPT). Blocks of letters (trials) are displayed on the screen, one after the other. Participants must press a key every time the correct stim-ulus appears (e.g., an A after an X). All other letters are distractors. Parameters recorded: R.T. (ms) and number of correct responses.

Episodic memory.

Test 4. Verbal Memory. The participant listens to the same series of words three times and, at the end of each sequence, must repeat aloud all the words he/she remembers. Parameters recorded: correct responses.

Verbal memory (delayed and recognition trials). In the delayed test, twenty minutes after the end of the Verbal Memory test, the participant must repeat aloud all the words he/she remembers from the list that was presented in that test. In the recognition test, immediately after the previous test, the participant listens to a group of words, half of which belong to the list presented in the Verbal Memory test, and must answer whether each of them was on that list. Parameters recorded: correct responses.

Test 5. Visual memory. Drawings of common objects are presented and the partici-pant must then repeat aloud all the drawings he/she remembers. Parameters recorded: correct responses.

Visual memory (delayed and recognition trials). In the delayed test, twenty minutes after the end of the Visual Memory test, the child has to repeat aloud all the pictures he/she remembers from the list that was presented in the test. In the recognition test, immediately after the previous test, images are presented, many of which appeared in the Visual Memory test, and for each of them the person must indicate whether it was in the sequence of the mentioned task. Parameters recorded: correct answers.

Language

Test 6. Verbal comprehension (images). A set of pictures, of a given category (e.g., animals), is shown and the participant is instructed (aurally) to select a picture that fulfils the indicated conditions (type of animal, position, activity it can perform and/or colour: e.g., "touch the frog next to the dog"). Parameters recorded: correct answers.

Test 7. Verbal comprehension (figures). Similar to the previous test, but with pictures of geometric figures (circles, triangles and squares: small, medium and large) of different colours, pointing out those that meet the indicated conditions (figure, size, position and/or colour: e.g. "touch a small blue circle"). Parameters recorded: correct answers.

Test 8. Phonetic fluency. A letter is indicated and the participant has to mention all the words he/she knows that begin with that letter. Parameters recorded: correct answers. Time: 60 seconds.

Executive function

Test 9. Working memory. After hearing sequences of mixed numbers and colours, the participant has to repeat the numbers and colours stated; first the numbers, in ascending order, and then the colours, or vice versa. Parameters recorded: correct answers.

Test 10. Abstract reasoning. A set of logical series is presented on screen. The par-ticipant must select the element that completes the series currently displayed. Parameters recorded: R.T. (ms) and correct answers.

Test 11. Semantic fluency. The participant listens to a semantic category (colours or animals), and must say all the items he/she knows from that category. Parameters rec-orded: correct answers. Time: 60 seconds.

Test 12. Inhibition: Go-No go. Two items are alternately displayed on the screen. In the first phase, the participant must press a key when one of them appears, and, after hearing a sound, which represents the change to the second phase or half of the test, press a key when the other item appears. Parameters recorded: T.R. (ms) and correct responses.

Test 13. Flexibility and inhibitory control: Spatial Stroop. A sequence of arrows is presented on the screen which can point to the left or to the right. Each time an arrow appears, the participant must press the LEFT ARROW key, if the arrow is pointing to the left or the RIGHT ARROW key if the arrow is pointing to the right. On some occasions, the stimuli are presented on the side of the screen consistent with the direction of the arrow, and on other occasions, the opposite. Parameters recorded: T.R. (ms), correct answers, errors by omission and commission

Test 14. Planning. An amusement park is presented and the role of the participant is to go on the maximum number of rides with the money provided. He/she will have to bear in mind that there is a time limit for being at the funfair and each ride takes up a part of that time. The objective of the task is that the participant manages to go on the highest number of different rides with the resources available. Parameters recorded: number of rides and number of different rides used.

  1. Finally, I wondered who did translate the tasks, please clarify. Why were the tasks chosen as they were, please motivate (maybe some former study did use the exact same tasks in a different sample, maybe those tasks measure abilities most critical for successful neurotypical development or pathologies).

We thank the reviewer for this comment and understand the difficulty of testing the usefulness of the battery without including more information on the contents and variables of the tests included.

On the one hand, the design of the battery was based on previous literature on neuropsychological assessment. On the other hand, the battery has been validated in Ecuadorian and Moroccan samples. We also have data that it discriminates between normotypic and preterm children (Spanish sample), and with different SES (Ecuadorian sample).

We have expanded the information in the method section of the manuscript and also provide additional information in the response to the reviewer.

The original version of the BENCI battery (Cruz-Quintana et al., 2013) was written in Spanish and was developed among Ecuadorian children as a result of an International Cooperative Program (A3/042954/11). BENCI is composed of basic neuropsychological domains required to conduct a comprehensive neuropsychological assessment (Lezak, Howieson, & Loring, 2004): speed processing, visuomotor coordination, attention, memory, language, and executive functions. Neuropsychological tests were developed to cover these domains. All these tests were developed using valid neuropsychological procedures based on the literature of neuropsychological assessment (Delis, Kramer, Kaplan, & Ober, 1987; Rey, 1964). 

As a result, the BENCI consists of 14 neuropsychological tests. This version has norms for Ecuadorian children between 6 and 11 years old and it has demonstrated good psychometric properties. The test-retest reliability was good (correlation ranged between r = .927 and r = .351). The convergent validity of the BENCI was calculated using a number of valid tests such as the Stroop Color Word test (Golden, 2001), Backward Digits (Woodcock & Sandoval, 1996), Raven´s Progressive Matrices (Raven, 1977), CCTT (Llorente et al., 2003), and the Spanish adaptation of the California Verbal Learning Test (TAVECI: Pamos et al., 2007). Results showed acceptable correlations between the BENCI subtest and these tests.

Furthermore, BENCI has demonstrated discriminant validity in a study that compared the neuropsychological performance of preterm children and normal children in Spanish sample (M= 6.7 years and SD = 0.614) (García Bermúdez et al., 2012).

In addition to the good psychometric properties of the BENCI battery, it is computerized but requires no previous experience to be administered. This computerized format allows for standardized administration, records hits, errors, or RT (when proceeded) in a very easy manner, and it is simple and enjoyable for children.

The Arabic version of the BENCI battery is also available (Cruz-Quintana et al., 2013). Translation and backtranslation were completed. One bilingual neuropsychologist translated the test from Spanish to Arabic and another bilingual neuropsychologist translated the BENCI from Arabic to Spanish. Adaptations to Arabic culture were needed for some words of the memory list and for some of the pictures (e.g., we changed the pig’s picture to a sheep).

We did not need to adapt numbers since Arabic numbers are used in Morocco. This battery has been translated and adapted according to the international Test Commission Guidelines and the Standards for Educational and Psychological Testing (American Educational Research Association, American Psychological Association, & National Council on Measurement in Education, 1999; International Test Commission, 2010).

  1. Next, given the long introduction on the impact of SES on EF performance and neurotypical development I wondered why the scores were analysed only including age and not SES, maybe the sample was to homogenous SES-wise.

In the introduction, reference is made to the SES as an example of the usefulness of the application of the BENCI battery in Ecuadorian samples. In this case, we understand that it is a measure of discriminant validation for the Ecuadorian sample, although it has not been the only measure of validity. To test the validity of the BENCI, both in the Ecuadorian sample and in the Moroccan sample, neuropsychological assessment tests have been used to calculate convergent validation, following a procedure similar to the one described in this paper and which have been analysed by the team of authors (Buerno-Garcés et al., 2019 and Fasfous et al., 2015). The results presented in this paper are validation results for a sub-sample of children aged 9 to 11 years old.

  1. In addition, the five-factor structure of the BENCI is also mentioned quite often without reference or fit indices which is deficient. So please provide at least the fit indices. However, this brings me to my next point, how was sample size determined and which age groups would be more informative (as suggested in the GD). Seventy-three seems a bit too small sample for a CFA. Please convince me otherwise.

There is no theoretical model that explains the structure of the entire battery, although we did rely on the review of previous literature on child neuropsychological assessment. However, we followed Diamond's (2013) 5-factor model to decide which tests to include for assessing executive functions.  The confirmatory factorial analysis results indicate that five factors exist: Inhibition, Flexibility, Fluency, Reasoning, and Working Memory. This five-factor model demonstrates that the chi-square analysis was not statistically significant. The CMIN/DF ratio (chi square/degrees of freedom) of the model was 1.240 which indicates a good adjustment. With respect to the global adjustment indicators, the CFI = 0.939 and the TLI = 0.924 demonstrate that it is an acceptable model, and by the RMSEA = 0.042, also indicates an excellent adjustment. These results suggest that it is a strong adjustment model.

On the other hand, we agree with the reviewer that the sample is small. However, the initial results of the Ecuadorian version only include children aged 6-11 years (Burneo-Garcés et al., 2019) and the Arabic version 7, 9 and 11 years (Fasfous et al., 2015). The equal distribution by age and gender has been controlled.

Thanks to other projects, we are extending the Spanish sample to include other age groups, as well as carrying out validation in other countries (Cuba). As previously mentioned, the BENCI was designed in Spain, and is already being used in other countries where validation data are available. We believe that it is necessary to inform the existence of validation data in a Spanish sample so that education and health professionals are aware of the battery. Because of its characteristics (easy to use and correct, free of charge, collects broad neuropsychological domains, provides specific data for each function assessed), the BENCI can be used to provide results of great value for health and education professionals. We have expanded the information included in the manuscript to justify the publication of validation data at ages 9, 10 and 11.

“Childhood is the most important stage of human development, as it is in the first years of life that the maturational and neurological foundations of development are laid. What, how and how much children learn in school depends largely on the social, emotional and cognitive competencies they have developed in their early years. Comprehensive instruments to assess children's neuropsychological performance are needed to identify children with possible deficits early, as well as to evaluate the effect of early intervention and prevention programs. The identification or diagnosis of potentially affected neurodevelopmental processes is one of the basic prevention activities. Neuropsychological assessment in children has specific characteristics since functional systems of the brain are assessed when development is not yet complete. Neuropsychological assessment of children requires specific tests to identify the child's performance in different neuropsychological domains. Some tests exist but, for the most part, they do not allow an initial screening, do not identify specific areas of deficit, but offer general scores and are expensive, which makes their use difficult, even more so in certain contexts. Taking these difficulties into account, it is necessary to have assessment instruments that are flexible, that can be modified according to age, that are attractive and quick to apply, and that have been developed specifically for children.”

  1. Thus, overall, I agree that test batteries are important, but I think they should be properly documented when validation studies are published. Thus, in a revision much more information needs to be provided to motivate readers to use the BENCI.

We agree with the point of view of the reviewer and we have included additional information about the BENCI battery in the manuscript.

Reviewer 2 Report

The Studies of the neurodevelopment of children are vital to promote a good quality of life during childhood. Few test batteries with adequate reliability and validity indices are available to evaluate the different neuropsychological domains.

The authors aimed to obtain initial evidence on the validity of the BENCI neuropsychological test battery in a Spanish population.

The authors administered to 73 children aged 9, 10, and 11 years. BENCI, TMT, Toni-2, Token, and WISC-IV Authors detected: (a) Significant differences among age groups were observed in the domains of sustained attention, memory, and executive function. (b) Significant correlations were found between BENCI subtests and the other neuropsychological tests. (c) Further research is warranted on the relationship of BENCI with other tests in wider age groups.

The authors concluded that their study  demonstrates that the Spanish version of BENCI has adequate validity to be used for evaluating the main neuropsychological domains in children.

The study is interesting.

Some minor comments for the authors:

1.       Avoid unsolved acronyms in the abstract.

2.       Aim “the objective of the present study was to obtain preliminary evidence on the validity”. Please be more explicit. What do you mean for preliminary evidence?

3.       Insert a table with acronyms.

4.       Methods. There are strange paragraphs. See “->Computerized Battery for Neuropsychological Evaluation of Children-BENCI [2].” Please write them according the MDPI style.

5.       The cocnlusions are very short and they must better match with the aim. The aim that I asked to correct was “to obtain preliminary evidence…” The conclusions “In conclusion, this study demonstrates that the Spanish version of BENCI has adequate validity to be used for evaluating the main neuropsychological domains in children” seems to demonstrate something more important than the aim.

6.       Add something, perhaps in the conclusion, on the further work to do.

7.       My language is not English, however I’d avoid the excessive use of “passive” verbs as in the abstract.

Author Response

Reviewer 2

The Studies of the neurodevelopment of children are vital to promote a good quality of life during childhood. Few test batteries with adequate reliability and validity indices are available to evaluate the different neuropsychological domains. The authors aimed to obtain initial evidence on the validity of the BENCI neuropsychological test battery in a Spanish population. The authors administered to 73 children aged 9, 10, and 11 years. BENCI, TMT, Toni-2, Token, and WISC-IV Authors detected: (a) Significant differences among age groups were observed in the domains of sustained attention, memory, and executive function. (b) Significant correlations were found between BENCI subtests and the other neuropsychological tests. (c) Further research is warranted on the relationship of BENCI with other tests in wider age groups. The authors concluded that their study demonstrates that the Spanish version of BENCI has adequate validity to be used for evaluating the main neuropsychological domains in children. The study is interesting.

Some minor comments for the authors:

  1. Avoid unsolved acronyms in the abstract.

We have re-written the abstract in order avoid the use of acronyms.

Study of the neurodevelopment of children is vital to promote a good quality of life during childhood. Few batteries showing adequate reliability and validity indices are available to evaluate the different neuropsychological domains. The objective of this study was to obtain initial evidence on the validity of the Computerized Battery for Neuropsychological Evaluation of Children (BENCI) in a Spanish population. For assessing validity the BENCI Battery and other measures of task switching, abstract reasoning, linguistic abilities, processing speed and attention were used. The sample was composed by a total of 73 children aged 9, 10, and 11 years. Significant differences among age groups were observed in the domains of sustained attention, memory, and executive function. In addition, the BENCI subtests showed statistically significant correlations with the other neuropsychological tools. Further research is warranted on the relationship of BENCI with other tests in wider age groups and to assess the factorial structure of the scale and the reliability values of the subtests. In conclusion, this study seems to indicate that the Spanish version of BENCI has promising validity evidences to be used for evaluating the main neuropsychological domains in children.

  1. Aim “the objective of the present study was to obtain preliminary evidence on the validity”. Please be more explicit. What do you mean for preliminary evidence?

Following the reviewer suggestion we have included the hypothesis in order to clarify the objective of the present research.

With this background, the objective of the present study was to obtain preliminary evidence on the validity of BENCI in a Spanish population, exploring its discriminant validity by comparing scores among different age groups and examining its convergent validity by comparing scores with the results of neuropsychological tests habitually employed to evaluate children. The hypothesis were: (1) BENCI tasks will discriminate between age groups with older children having better scores than younger children and (2) positive correlations will be found between BENCI tasks and other neuropsychological test that assess the same domain (task switching, abstract reasoning, linguistic abilities, processing speed and attention) 

In this manuscript we present data from the validation of the BENCI in a Spanish sample aged between 9 and 11 years old. The equal distribution by age and gender has been controlled for. However, thanks to other projects, we are extending the Spanish sample to include other age groups, as well as carrying out validation in other countries (Cuba). As previously mentioned, the BENCI is designed in Spain, and is already being used in other countries where validation data are available. We believe that it is necessary to inform the existence of validation data in a Spanish sample so that education and health professionals are aware of the battery.  Because of its characteristics (easy to use and correct, free of charge, collects broad neuropsychological domains, provides specific data for each function assessed), the BENCI can be used to provide results of great value for health and education professionals. We have expanded the information included in the manuscript to justify the publication of validation data at ages 9, 10 and 11.We have expanded the information included in the manuscript to justify the publication of validation data at ages 9, 10 and 11. However, the initial results of the Ecuadorian version only include children aged 6-11 years (Burneo-Garcés et al., 2019) and the Arabic version 7, 9 and 11 years (Fasfous et al., 2015).

  1. Insert a table with acronyms.

Following the reviewer suggestion we have included in the different tables of the manuscript a description of each of the acronyms used. We think that this information will be more useful that using a table for acronyms.

  1. Methods. There are strange paragraphs. See “->Computerized Battery for Neuropsychological Evaluation of Children-BENCI [2].” Please write them according the MDPI style.

Thank you for the commentary. We have modified the Method section in order to better describe the BENCI Battery and we have modified the rest of the paragraphs according to the MDPI style.

  1. The conclusions are very short and they must better match with the aim. The aim that I asked to correct was “to obtain preliminary evidence…” The conclusions “In conclusion, this study demonstrates that the Spanish version of BENCI has adequate validity to be used for evaluating the main neuropsychological domains in children” seems to demonstrate something more important than the aim.

Thank you for your comment. We have added the following information to the discussion and in the conclusion section.                                                                                                        

Childhood is the most important stage of human development, as it is in the first years of life that the maturational and neurological foundations of development are laid. What, how and how much children learn in school depends largely on the social, emo-tional and cognitive competencies they have developed in their early years. Comprehen-sive instruments to assess children's neuropsychological performance are needed to iden-tify children with possible deficits early, as well as to evaluate the effect of early interven-tion and prevention programmes. The identification or diagnosis of potentially affected neurodevelopmental processes is one of the basic prevention activities. Neuropsychologi-cal assessment in children has specific characteristics since functional systems of the brain are assessed when development is not yet complete. Neuropsychological assess-ment of children requires specific tests to identify the child's performance in different neuropsychological domains. Some tests exist but, for the most part, they do not allow an initial screening, do not identify specific areas of deficit, but offer general scores and are expensive, which makes their use difficult, even more so in certain contexts. Taking these difficulties into account, it is necessary to have assessment instruments that are flexible, that can be modified according to age, that are attractive and quick to apply, and that have been developed specifically for children.

In conclusion, this study demonstrates that the Spanish version of BENCI has ade-quate validity to be used for evaluating the main neuropsychological domains in children. Adequate discriminant validity, showing that older children tend to have higher scores in some of the BENCI test was shown. In addition, positive and significant correlations were found between neuropsychological tests and BENCI tests that targeted the same neuropsychological domain. These results should be taken with caution and further research is needed with larger sample sizes that allows for testing the factorial structure of the BENCI in Spanish population of children and adolescents.

  1. Add something, perhaps in the conclusion, on the further work to do.

Following the reviewer’s suggestion we have included additional information in the conclusion section of the manuscript.

In conclusion, this study demonstrates that the Spanish version of BENCI has ade-quate validity to be used for evaluating the main neuropsychological domains in children. Adequate discriminant validity, showing that older children tend to have higher scores in some of the BENCI test was shown. In addition, positive and significant correlations were found between neuropsychological tests and BENCI tests that targeted the same neuropsychological domain. These results should be taken with caution and further research is needed with larger sample sizes that allows for testing the factorial structure of the BENCI in Spanish population of children and adolescents.

  1. My language is not English, however I’d avoid the excessive use of “passive” verbs as in the abstract.

We have modified the abstract section and changed the tense of the sentences following your suggestion.

Study of the neurodevelopment of children is vital to promote a good quality of life during childhood. Few batteries showing adequate reliability and validity indices are available to evaluate the different neuropsychological domains. The objective of this study was to obtain initial evidence on the validity of the Computerized Battery for Neuropsychological Evaluation of Children (BENCI) in a Spanish population. For assessing validity the BENCI Battery and other measures of task switching, abstract reasoning, linguistic abilities, processing speed and attention were used. The sample was composed by a total of 73 children aged 9, 10, and 11 years. Significant differences among age groups were observed in the domains of sustained attention, memory, and executive function. In addition, the BENCI subtests showed statistically significant correlations with the other neuropsychological tools. Further research is warranted on the relationship of BENCI with other tests in wider age groups and to assess the factorial structure of the scale and the reliability values of the subtests. In conclusion, this study seems to indicate that the Spanish version of BENCI has promising validity evidences to be used for evaluating the main neuropsychological domains in children.

Round 2

Reviewer 1 Report

thanks for taking into consideration all of my comments and good luck with your ongoing project.

This manuscript is a resubmission of an earlier submission. The following is a list of the peer review reports and author responses from that submission.